# Contrastive Learning of Global-Local Video Representations

**Shuang Ma**
Microsoft
Redmond, WA, USA

**Zhaoyang Zeng**
Sun Yat-sen University
Guangzhou, China

**Daniel McDuff**
Microsoft Research
Redmond, WA, USA

**Yale Song**
Microsoft Research
Redmond, WA, USA

## Abstract

Contrastive learning has delivered impressive results for various tasks in the self-supervised regime. However, existing approaches optimize for learning representations specific to downstream scenarios, i.e., *global* representations suitable for tasks such as classification or *local* representations for tasks such as detection and localization. While they produce satisfactory results in the intended downstream scenarios, they often fail to generalize to tasks that they were not originally designed for. In this work, we propose to learn video representations that generalize to both the tasks which require global semantic information (e.g., classification) and the tasks that require local fine-grained spatio-temporal information (e.g., localization). We achieve this by optimizing two contrastive objectives that together encourage our model to learn global-local visual information given audio signals. We show that the two objectives mutually improve the generalizability of the learned global-local representations, significantly outperforming their disjointly learned counterparts. We demonstrate our approach on various tasks including action/sound classification, lip reading, deepfake detection, event and sound localization.[1]

## 1 Introduction

Recent years have seen a surge of interest in contrastive self-supervised learning (CSL) [59, 38, 35, 16] to obtain representations that generalize to various downstream scenarios. In CSL, the choice of "contrasting views" plays a crucial role because the learned representations capture information shared between different views by maximizing mutual information between them [8]. This makes it critical to design contrastive objectives with the "right" contrasting views tailored for the intended downstream scenarios [77], which has been the focus of many recent works [38, 10, 45, 74, 84, 83].

The progress made so far provides important insights for understanding how to select optimal contrasting views for a given task [77]. However, the current paradigm of designing CSL approaches specific to any intended (global or local) downstream scenarios could be suboptimal, as in the real-world case the downstream scenarios are generally unknown in advance. This not only limits the generalizability of the learned representations, evaluating the approaches solely on the intended scenarios could produce misleading conclusions. Although existing approaches achieve impressive results in their intended downstream tasks, they often fail to generalize to tasks that they were not originally designed for, e.g., as we show later in our experiments, global representations do not generalize well to tasks such as lip reading [22, 21] which require local spatio-temporal information.

Motivated by this, we take an orthogonal direction to the current CSL approaches: We aim to learn representations agnostic to the types of downstream scenarios and generalize to both the scenarios that require global representations (e.g., classification) and scenarios that require local representations (e.g., localization). We focus on learning video representations using the natural

---

[1] https://github.com/yunyikristy/global_local

35th Conference on Neural Information Processing Systems (NeurIPS 2021).

audio-visual correspondence as the primary self-supervisory signal. In this scenario, the notion of global/local representations is intertwined in space and time; we can obtain representations that are *spatially* global or local, and also *temporally* global or local. However, most existing video CSL approaches optimize for only global spatio-temporal representations and demonstrate them on audio/visual video classification tasks [60, 46, 55, 61]. Part of the difficulty here is that formulating a contrastive objective for local representations is not straightforward because of the one-to-many relationship in audio-visual correspondences, i.e., spatially, multiple pixel regions can contribute to the sound in the corresponding audio, and temporally, multiple temporal slices of audio can map to a single video frame due to sampling rate differences. This hinders the development of CSL for local video representations useful for tasks such as sound source separation and lipreading.

In this paper, we present an approach for learning global-local video representations in the CSL framework. We design two cross-modal contrastive objectives that collaboratively capture information shared between audio and visual signals. An important aspect of our approach is the factorization of the spatio-temporal feature space into a *spatially-local/temporally-global* subspace and a *spatially-global/temporally-local* subspace, where each of the two contrastive objectives are defined in, respectively; see Fig. 1. The explicit space-time factorization helps each contrastive objective focus on capturing either spatially-local or temporally-local information and thus facilitates learning complementary features from audio-visual correspondence more effectively than in the original spatio-temporal space. Furthermore, we define both objectives in the multiple instance learning framework [24, 52] to handle the one-to-many relationship between audio and visual signals. This helps the model learn representations without knowing fine-grained audio-visual correspondence.

We evaluate our approach on various downstream tasks that need *local* spatio-temporal information, i.e., lip reading [22, 21, 3], deep-fake detection [25] and audio-visual event localization [76], and also discriminative tasks that needs *global* information, i.e., audio/visual video classification [71, 47, 63, 41]. We show that the same pretrained model successfully generalizes to all our scenarios without having to re-pretrain it using different objectives and/or datasets. Furthermore, we demonstrate that the two contrastive objectives mutually benefits each other and helps improve the generalizability of both global and local representations. To the best of our knowledge, our work is the first to demonstrate a CSL approach that learns video representations that generalize to both global and local video understanding scenarios.

## 2 Related Work

**Contrastive self-supervised learning.** Contrastive learning leverages multiple views of the same data [59], e.g., multiple perspectives *within the same modality* such as augmentations of the same image, different frames/clips of a video, etc. [35, 37, 32] or perspectives from *different modalities* such as RGB and depth, images/videos and text [75, 73, 53, 4]. DIM [38] and SimCLR [16] show that leveraging local information in contrastive learning further improves performance on image classification. DIM [38] has been extended to multi-scale [11] and to video data [37]. However, evaluation is still focused on "discriminative" tasks, e.g., image classification and video classification, while there is little evidence that these models will adapt well to tasks that require local information.

Several recent advances happened in the image domain, e.g., MoCo [35, 17, 19], BYOL [31], SwAV [13], SimSiam [18], BarlowTwins [90]. Although evaluation is performed in both global and local downstream scenarios such as image classification, object detection, semantic/instance segmentation and depth estimation, this line of work focuses on evaluating generalizability of the learned global representations rather than studying the importance of global and local information in generalization, which is the focus of this work. They also focus on image recognition tasks only.

**Audio-visual video representation learning.** Learning video representations from the natural audio-visual correspondence has been studied extensively. Most existing approaches aim to capture high-level semantic information useful for sequence-level (global) discrimination tasks such as audio/visual video classification [9, 46, 5, 55, 61, 23]. Along this line of work, AVSlowFast [81] utilizes different temporal scales of the audio and visual data, which encourages the model to capture fine-grained temporal information. However, their learning objective optimizes for only *global* representations induced by different sampling rates and their evaluation is still focused on classification tasks.

Another line of work focuses on capturing fine-grained spatio-temporal local information suitable for "local" task such as sound source separation and localization [7, 69, 93, 92, 28, 88, 64, 50]. Lin et

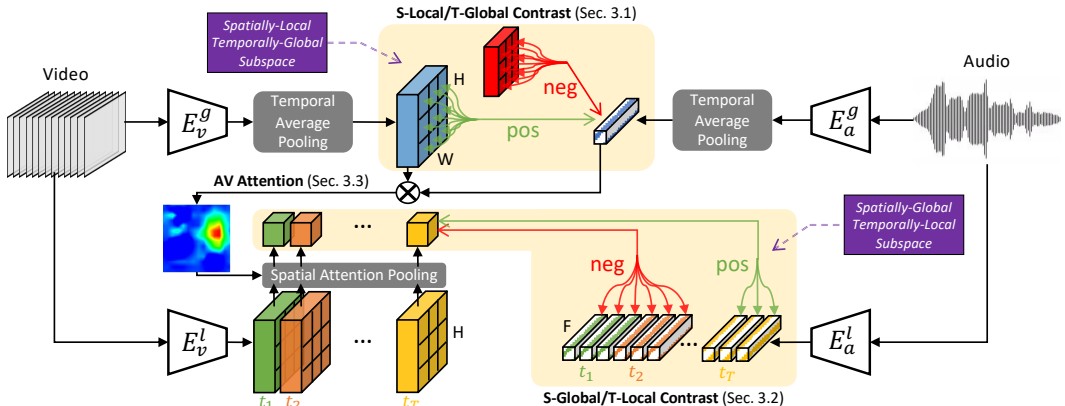

Figure 1: We factorize the joint spatio-temporal feature space into a spatially-local/temporally-global subspace and a spatially-global/temporally-local subspace. In each space we define a cross-modal contrastive objective that compare audio-visual signals in the multiple instance learning framework. For notational simplicity we indicate only the temporal scale in the superscripts of all the variables.

al. [50] learn patch-level audio-visual correspondence by drawing positive/negative patches iteratively from sounding/non-sounding regions induced by audio-visual feature correlation. However, their approach learns spatially-local/temporally-global representations only, and their evaluation is focused on localization tasks. In contrast to these previous work, we explicitly optimize for global and local spatio-temporal representations and evaluate on both classification and localization downstream tasks. There are also some work proposed to learn audio-visual representations, however they are specifically designed for a particular task, e.g. speaker recognition [56]. While our work learns general-purpose video representation and is demonstrated on a variety of downstream tasks.

## 3 Approach

We factorize the feature space into a *spatially-local/temporally-global (S-local/T-global)* subspace and a *spatially-global/temporally-local (S-global/T-local)* subspace and define two cross-modal contrastive objectives in each of the subspaces. The S-local/T-global objective captures slowly changing patch-level information (Sec.3.1) while the S-global/T-local objective captures fast changing frame-level information (Sec.3.2). Furthermore, we utilize the learned patch-level information to guide learning frame-level information via a spatially-aware attention pooling mechanism (Sec.3.3).

### 3.1 Spatially-Local Temporally-Global Contrastive Objective

The purpose of this objective is to capture slowly changing patch-level information with high audio-visual correlation. We define an audio encoder $E_a^g$ that computes audio embeddings $z_a^g \in \mathbb{R}^{T_a \times F}$, where $T_a$ is the audio sequence length and $F$ is the number of frequency bands, and perform temporal average pooling to obtain $z_a^g \in \mathbb{R}^{1 \times F}$. Similarly, we define a visual encoder $E_v^g$ that computes visual embeddings $z_v^g \in \mathbb{R}^{T_v \times H \times W \times C}$, where $T_v$ is the visual sequence length and $H$, $W$, $C$ are the height, width and channel dimensions, and perform temporal average pooling to obtain $z_v^g \in \mathbb{R}^{1 \times H \times W \times C}$.

To help our model capture slowly changing yet spatially-detailed information, we purposely make input frames lack local temporal information by feeding them at a low sampling rate, e.g., at one eighths of the original sampling rate, which has been shown to be effective at capturing fine-grained spatial details [27]. For audio, we found that keeping the original sampling rate to be more effective.

We consider $z_a^g$ and $z_v^g$ that come from the same clip as positive pairs, while features coming from different clips are negatives. To capture local *spatial* information in $z_v^g$, we consider each cell of the $H \times W$ spatial grid as an instance and define pairs by coupling $z_a^g$ with $z_v^g[i], \forall i \in H \times W$. Note that not all $z_v^g[i]$ will have valid audio-visual correspondence to $z_a^g$; it is rather likely that only a few cells (e.g., the lip region of a talking person) will match the information in $z_a^g$. To resolve this

misalignment issue, we define our contrastive loss in the multiple instance learning framework [53]:

$$\mathcal{L}^g = -\log\left(\frac{\sum_{z_v^g[i]\in\mathcal{P}} F(z_a^g, z_v^g[i])}{\sum_{z_v^g[i]\in\mathcal{P}} F(z_a^g, z_v^g[i]) + \sum_{z_v'\in\mathcal{N}} F(z_a^g, z_v'^g)}\right) \tag{1}$$

where $F(z_a, z_v) = \exp(z_a^T z_v)$ measures the compatibility between $z_a$ and $z_v$, $\mathcal{P}$ is a set of spatial grids in $z_v^g$, and $\mathcal{N}$ is a set of negative visual instances taken from different clips, i.e., given a batch of $B$ video clips, we consider $H \times W \times (B-1)^2$ negative pairs.

### 3.2 Spatially-Global Temporally-Local Contrastive Objective

To capture information sensitive to temporal changes, e.g., the lip region of a talking head, we take a temporally-granular approach and contrast visual features at each time step to the corresponding audio features in a local temporal neighborhood. We compute audio embeddings $z_a^l \in \mathbb{R}^{T_a \times F}$ and visual embeddings $z_v^l \in \mathbb{R}^{T_v \times H \times W \times C}$ in the same manner as before but without applying temporal average pooling at the end. To help our model capture fine-grained temporal changes, unlike in the previous objective, we feed frames at a higher sampling rate and perform spatial pooling over visual embeddings to obtain $z_v^l \in \mathbb{R}^{T_v \times 1 \times 1 \times C}$.

We construct contrasting views by taking audio and visual features from the same local temporal block as positive pairs; audio-visual pairs from different temporal blocks of the *same* video become negatives. We do not use samples from different videos in order to encourage the shared information between modalities to be just *how features vary over time*.

Note that a single visual feature $z_v^l[t], t \in T_v$ maps to multiple audio features $z_a^l[t], t \in T_a$ because audio signals are typically captured at a higher sampling rate. Therefore, unlike in the previous objective, all the pairs within the same local temporal block can be considered as valid positives. There exists several ways to define a contrastive objective with multiple positives [42, 12, 70]; we opt for a simple approach that sums scores over all positive pairs, which leads to a form similar to Eqn. 1:

$$\mathcal{L}^l = -\log\left(\frac{\sum_{z_a^l[t_k]\in\mathcal{P}} F(z_a^l[t_k], z_v^l[t])}{\sum_{z_a^l[t_k]\in\mathcal{P}} F(z_a^l[t_k], z_v^l[t]) + \sum_{z_a'\in\mathcal{N}} F(z_a', z_v^l)}\right) \tag{2}$$

$\mathcal{P}$ is a set of audio features within the same temporal block of $z_v^l[t]$ and $\mathcal{N}$ is a set of all possible audio-visual slice pairs that come from different parts of the same video. Given a video with $T_v$ frames and $T_a = M \times T_v$ audio slices, we consider $M \times (T_v - 1)^2$ negative pairs for a video sequence.

### 3.3 Spatial Pooling with Audio-Visual Attention

To define the two subspaces we apply global pooling over either spatial or temporal dimensions. For the S-local/T-global subspace, we simply perform temporal average pooling because video inputs to our model are relatively of short length (3 seconds) and tend to show content recorded in a single scene. For the S-global/T-local subspace, however, spatial average pooling could be problematic because not all pixel regions will contribute to the sound in the corresponding audio signals.

Therefore, we use the learned S-local/T-global representation (Sec. 3.1) to obtain an audio-visual attention map and use it to perform spatial attention pooling, as seen in Fig. 1. Specifically, we compute the dot product between $z_a^g$ and each of $z_v^g[i], \forall i \in H \times W$ to obtain an attention map indicating regions of high audio-visual correlation. We use this map to perform spatial pooling at each time step to obtain $z_v^l[t], \forall t \in T_v$. We demonstrate the effectiveness of spatial attention pooling in Table 8 and show qualitative examples in Fig. 2 for the scenario of sound source localization.

## 4 Experiments

**Implementation details.** We use 3D-ResNet [34] for our visual encoders ($E_v^g$ and $E_v^l$) and 1D-ResNet [36] for our audio encoders ($E_a^g$ and $E_a^l$), in both cases using Batch Normalization [39]. We share the weights of the two audio encoders and denote both simply by $E_a$; for the visual encoders we instead keep the weights separate in order to encourage them to capture complementary information from two different sampling rates; this can be seen as an instantiation of the SlowFast network with no

lateral connection [27]. All models are trained end-to-end using ADAM [44] with an initial learning rate $\gamma = 10^{-3}$ after a warm-up period of 500 iterations. We use 16 NVIDIA Tesla P100 GPUs with a batch size of 32. For the S-local/T-global objective we use features at a $16 \times 16$ spatial resolution. To compute the S-global/T-local objective, we adopt a temporal window of size three without overlap.

During pretraining we sample frames at 10 FPS and apply random cropping, horizontal flipping, gray-scaling, and temporal jittering. We set the clip length to 32 frames (3 seconds) and resize frames to $112 \times 112$; we feed 8 frames and 32 frames to $E_v^g$ and $E_v^l$, respectively. We extract mel-spectrograms from the raw waveform using LibROSA and get a $80 \times T$ matrix with 80 frequency bands; T is proportional to the length of an audio clip. We then segment the mel-spectrogram according to the corresponding video clips to ensure temporal alignment. We treat the mel-spectrograms as an 80-channel 1D signal. For downstream tasks we follow the standard data preprocessing protocols.

**Datasets.** Many video tasks involve human actions (e.g., action recognition), faces (e.g., deepfake), and speech (e.g., lip reading). Pretraining our model on different datasets for different downstream tasks could produce misleading conclusions as the model could simply pick up the biases in the data useful for downstream tasks, e.g., Kinetics [14] contains various human actions useful for solving action recognition and AVSpeech [26] mostly contains human speech useful for lip reading, but not the other way around. Therefore, we pretrain our model *only once for all* downstream tasks using a combination of Kinetics [14] and AVSpeech [26]. The standard choice of video dataset for pretraining is Kinetics-400 [41] that contains 240K videos. We match the size by randomly selecting 120K video samples from each datasets; we term it as K-AV-240K. For the ablation study, we pretrain our model on a subset of 15K samples from the K-AV-240K dataset. For fair comparisons with existing work, we also pretrain our model on the same datasets as with state-of-the-art (SOTA) approaches.

We evaluate our pretrained model on action recognition (UCF101 [71], HMDB51 [47], and Kinetics400 [41]), on sound classification (ESC50 [63]), on lip reading (LRW [22] and LRS2 [21]), on deepfake detection (DFDC [25]), and on audio-visual event localization (AVE [76]). We also conduct qualitative analysis on sound source separation on Kinetics-Sounds [6].

## 4.1 Downstream Scenarios

**Lip Reading**. Visually recognizing a speaker's utterance is a challenging task, e.g., lip movements for different sounding letters can be visually similar to each other (e.g., /b and /p/, /d and /t). This requires visual representations to capture fine-grained spatio-temporal information. For a fair comparison with SOTA, we use the standard data processing protocol of [91]. We detect 68 facial landmarks in each frame using dlib [15] and use the outer eye and nose tip landmarks to align the detected face in each frame using an affine transform. Finally, an image of size $112 \times 112$ is cropped from the aligned face with the lip landmarks at the center, so that the lip region occupies one third of the image width. We apply random horizontal flipping as data augmentation. We concatenate the features produced by our pretrained $E_v^g$ and $E_v^l$ and feed them to a 2-layer MLP prediction head. For LRS2, we apply spatial average pooling before concatenation to preserve the temporal dimension and train the model using the CTC loss [30]. For LRW we apply spatio-temporal average pooling before concatenation and train the model on the cross-entropy loss. In both cases we train the whole model end-to-end.

Table 1 compares our approach with SOTA supervised and self-supervised methods. For LRS2, we report the word error rate (WER; the lower the better); for LRW, we evaluate on a 500-way word classification task and report top-1 accuracy (the higher the better). The results show that our approach with the same ResNet18 backbone outperforms SOTA supervised approaches on LRS and LRW by large margins, i.e. 4.7% WER reduction on LRS2 and 5.1% accuracy improvements on LRW. All the baseline self-supervised methods optimize for variants of global contrastive objectives and generally perform poorly on all three datasets. Our approach outperforms all SOTA self-supervised approaches with the same backbone and using the same pretraining dataset. These results demonstrate the importance of capturing fine-grained spatio-temporal information necessary for lip reading.

**Deepfake Detection**. We observe that "deepfakes" tend to be characterized by fine-grained audio-visual inconsistencies such as misalignment between lip motions and audio, unnatural facial and lip appearance/movements or asymmetry between facial regions such as the left and right eyes. Detecting such artifacts requires *local* spatio-temporal features. We take our pretrained model and finetune it on 1 second video clips from the DFDC dataset [25] for 100 epochs with a batch size of 16. We evaluate performance using video-wise Area Under the Curve (AUC). We follow the same data preprocessing

| Method | Backbone | Pretrained on | LRS2↓ | LRW↑ |
|---|---|---|---|---|
| WAS [21] | Conv. | N/A | 70.4 | 76.2 |
| STF [91] | ResNet18 | N/A | 51.7 | 83.7 |
| TM-CTC [2] | ResNet18 | N/A | 65.0 | - |
| TM-sep2seq [2] | ResNet18 | N/A | 49.8 | - |
| LRW [22] | VGG-M | N/A | - | 61.1 |
| Perfect Match [23] | TC-5 | N/A | - | 71.6 |
| ResNet-LSTM [72] | ResNet34 | N/A | - | 83.0 |
| TwoStream [79] | I3D | N/A | - | 84.1 |
| DFTN [82] | ResNet18 | N/A | - | 84.1 |
| MoCo [35] | ResNet18 | K-AV-15K | 71.5 | 61.2 |
| CPC [59] | ResNet18 | K-AV-15K | 66.7 | 65.3 |
| DPC [59] | ResNet18 | K-AV-15K | 65.1 | 67.5 |
| AVSlowFast [81] | ResNet18 | K-AV-15K | 56.1 | 75.8 |
| VDIM [37] | ResNet18 | K-AV-15K | 53.2 | 70.7 |
| | ResNet18 | K-AV-15K | 47.8 | 83.7 |
| Ours[2] | ResNet50 | K-AV-15K | 45.0 | 85.5 |
| | ResNet18 | K-AV-240K | **45.1** | **89.2** |

| Method | Backbone | Pretrained on | DFDC (AUC)↑ |
|---|---|---|---|
| Capsule [58] | VGG-19 | N/A | 53.3 |
| Multi-task [57] | Y-shape | N/A | 53.6 |
| HeadPose [89] | - | N/A | 55.9 |
| Two-stream [95] | Inception3 | N/A | 61.4 |
| Xception-c23 [67] | XCeption | N/A | 72.2 |
| Meso4 [1] | Inception4 | N/A | 75.3 |
| DSP-FWA [48] | - | N/A | 75.5 |
| Siamese [54] | - | N/A | 84.4[†] |
| MDS [20] | ResNet18 | N/A | 91.5[†] |
| MoCo [35] | ResNet18 | K-AV-15K | 60.2 |
| CPC [59] | ResNet18 | K-AV-15K | 67.9 |
| DPC [33] | ResNet18 | K-AV-15K | 71.2 |
| AVSlowFast [81] | ResNet18 | K-AV-15K | 80.9 |
| VDIM [37] | ResNet18 | K-AV-15K | 85.3 |
| | ResNet18 | K-AV-15K | 90.1 |
| Ours | ResNet50 | K-AV-15K | 92.7 |
| | ResNet18 | K-AV-240K | **96.7** |

Table 1: Comparison with SOTA on lipreading: LRS2 [21] (word error rate (WER); lower is better) and LRW [22] (top-1 accuracy; higher is better).

Table 2: Comparison with SOTA on deepfake detection [25]. †: [54, 20] use audio-visual signals; all the other methods use visual signals only.

| Method | Backbone | Pretrained on | UCF101↑ | HMDB51↑ | ESC50↑ |
|---|---|---|---|---|---|
| Random Forest [63] | MLP | N/A | - | - | 44.3 |
| ConvNet [62] | ConvNet-4 | N/A | - | - | 64.5 |
| ConvRBM [68] | ConvNet-4 | N/A | - | - | 86.5 |
| Scratch | ResNet18 | N/A | 46.5 | 17.1 | - |
| Supervised | ResNet18 | ImageNet | 82.8 | 46.7 | - |
| MotionPred [78] | C3D | K400-240K | 61.2 | 33.4 | - |
| RotNet3D [40] | ResNet18 | K400-240K | 62.9 | 33.7 | - |
| ST-Puzzle [43] | ResNet18 | K400-240K | 65.8 | 33.7 | - |
| ClipOrder [85] | R(2+1)D-18 | K400-240K | 72.4 | 30.9 | - |
| DPC [32] | ResNet34 | K400-240K | 75.7 | 35.7 | - |
| CBT [73] | S3D&BERT | K400-240K | 79.5 | 44.6 | - |
| SeLaVi [9] | R(2+1)D-18 | K400-240K | 83.1 | 47.1 | - |
| XDC [5] | R(2+1)D-18 | K400-240K | 84.2 | 47.1 | 78.0 |
| AVTS [46] | MC3 | K400-240K | 85.8 | 56.9 | 76.7 |
| AVID [55] | R(2+1)D-18 | K400-240K | 87.5 | 60.8 | 79.1 |
| GDT [61] | R(2+1)D-18 | K400-240K | 89.3 | 60.0 | - |
| Ours | ResNet18 | K-AV-240K | 90.1 | 61.3 | **80.1** |
| | ResNet18 | K400-240K | **91.1** | **61.9** | 79.8 |

Table 3: Comparison with SOTA on action classification (UCF101 [71], HMDB51 [47]) and sound classification (ESC50 [63]). We highlight the **best** results and the second best results.

protocol as in SOTA approaches for this task, and use the same training and test sets as [20]. We perform face detection to crop the face region in each video frame. We concatenate the features produced by our pretrained $E_v^g$ and $E_v^l$ after applying spatio-temporal average pooling, feed them to a 2-layer MLP prediction head, and train the whole network end-to-end using the cross-entropy loss.

Table 2 shows the results. Two of the baselines, [20] and [54], use both visual and audio features; all the other methods use only the visual features. We can see that when using only the visual features, our approach outperforms all previous SOTA approaches (AUC=96.7). We also compare our model with SOTA self-supervised approaches. Again, the baseline self-supervised methods perform poorly on this task which require local spatio-temporal information. Our model outperforms the best self-supervised result, highlighted in blue, by a large margin (90.1 vs. 85.3).

**Audio-Visual Event Localization.** An "audio-visual event" is defined as an event that is both visible and audible in a video segment. Audio-video event localization is usually evaluated in two settings, i.e. *fully-supervised* and *weakly-supervised*. The former aims to predict which temporal segment of a video has an audio-visual event and what category the event belongs to; the latter assumes that only a video-level event category is available and there is no temporal event boundary information during training. This is a challenging task because an event usually appears only in a small portion of

---

[2]Blue: comparisons of ours with the self-supervised approaches under the same setting. Underline: best reported results of supervised methods. All the supervised results are from the literature; all the self-supervised results are ours. N/A: models are trained from scratch on target datasets. ↑ / ↓: higher/lower is better.

frames within a video. Detecting which temporal segment contains an audio-visual event requires fine-grained ("local") spatio-temporal representations, especially in *weakly-supervised* setting, where there is no clue about temporal event boundaries.

In Table 4, we show comparisons with the SOTA on audio-visual event localization. Again, we take the same K-AV-240K-pretrained model used in the previous experiments and finetune it on the AVE dataset [76]. We concatenate the features produced by our pretrained $E_v^l$ and $E_a^l$, and feed them to a 2-layer MLP prediction head. Especially, we first utilize the $E_a^l$ to perform a spatial attention pooling on $E_v^l$ to highlight the "important" local spatial information. In this way, the final concatenated audio-visual features contain the desired the spatio-temporal local information. We evaluate our model on both *fully-supervised* (Fully-super.) and *weakly-*

| Method | Fully-super. | Weakly-super. |
|---|---|---|
| AVEL [76] | 68.6 | 66.7 |
| AVSDN [49] | 72.6 | 67.3 |
| CMAN [87] | 73.3 | 70.4 |
| DAM [80] | 74.5 | - |
| AVRB [66] | 74.8 | 68.9 |
| AVIN [65] | 75.2 | 69.4 |
| AVT [51] | 76.8 | 70.2 |
| CMRA [86] | 77.4 | 72.9 |
| PSP [94] | 77.8 | 73.5 |
| Ours | **82.1** | **79.8** |

Table 4: Comparison with SOTA on audio-visual event localization.

*supervised* (Weakly-super.) settings. For a fair comparison, we followed the same protocol and evaluation metric as [76]. Without bells and whistles, we achieved 82.1% localization accuracy on *fully-supervised* setting, and 79.8% on *weakly-supervised* setting, which all outperforms the SOTA, i.e. PSP [94] (77.8%) (Fully-super.) and 73.5% (weakly-super.), by large margins.

**Sound Source Localization.** To further demonstrate our approach achieving good audio-visual localization, we visualize audio-visual spatial attention maps on Kinetics-Sounds [6] (see Fig. 2), which contains videos deemed to have high audio-visual correspondence. Such visualization can also be considered as performing sound source localization, i.e. locate objects that making sound. To plot the figure, we use the learned audio-visual attention map, add a softmax layer and apply bilinear interpolation of the $16 \times 16$ attention map back to the original image size, i.e. $192 \times 192$. The figure shows that our learned attention maps successfully localize sounding sources in videos, especially when visual content is highly related to the corresponding audio signal. For example, the first row (video frames from "playing instruments") shows that our model can successfully localize the sounding region. For other activities like "baby talking," "playing basketball," "running," our model successfully highlights regions with humans. However, we find that the attention map incorrectly highlights regions on videos that have ambiguous audio-visual relation. We show example failure cases in the last two columns of the third row: There is no visual content that clearly relates with the audio signal, and thus the model fails to find sounding sources.

**Action and Sound Classification.** To evaluate the effectiveness of the learned global spatio-temporal representations, we evaluate our approach on action and sound classification. For action classification, we concatenate the visual features from $E_v^g$ and $E_v^l$ after spatio-temporal average pooling, feed them to a 2-layer MLP prediction head, and train the whole network end-to-end using the cross-entropy loss. For audio classification, we apply temporal average pooling to the audio features from $E_a$ and feed them to a 2-layer MLP prediction head, which is trained using the cross-entropy loss.

Table 3 shows the results. For a fair comparison to existing approaches, we report both the results pretrained on K-AV-240K and the results pretrained on Kinetics-400 [41] that contains 240K videos (K400-240K). We find that, while the K400-240K pretrained models leads to better performance on UCF101 and HMDB51 due to the similarities between datasets (all focus on human actions), the K-AV-240K pretrained models also achieve competitive results, outperforming all the baselines. Overall, on all three benchmarks, our approach achieves new SOTA results (91.1% on UCF101, 61.9% on HMDB51 and 80.1% on ESC50), demonstrating the effectiveness of learning global-local representations even for tasks that require global information. This suggests that optimizing for local representations also helps improve performance on classification tasks that require global information.

### 4.2 Ablation and Analysis

**Comparison of different contrastive objectives.** In Table 5 we compare various contrastive objectives on tasks that require fine-grained local spatio-temporal information. All the methods use

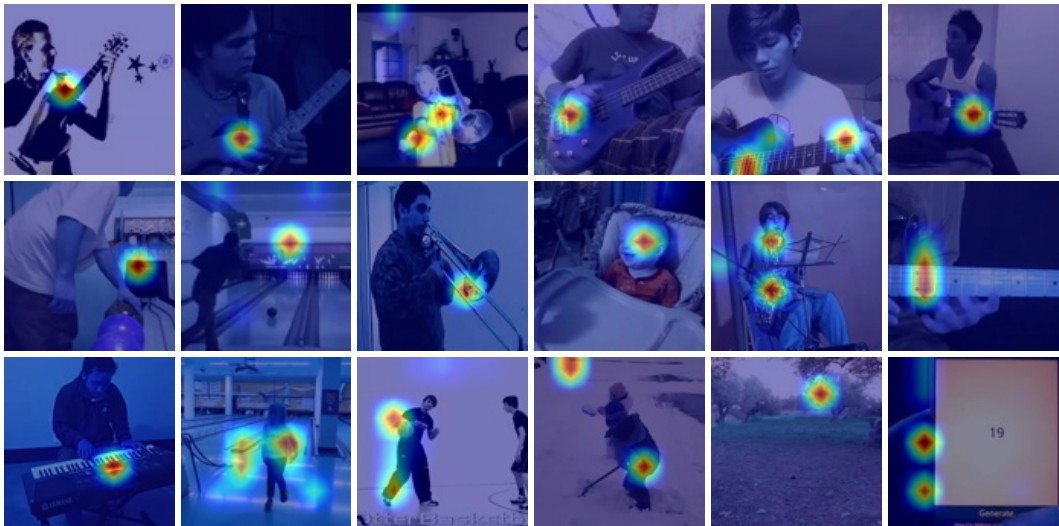

Figure 2: Visualization of the learned audio-visual spatial attention maps on videos from Kinetics-Sounds [6] show that they successfully locate sounding sources, e.g., musical instruments.

| Method | Contrastive Objective | X-Modal | Spa. | Temp. | LRS2↓ | LRW↑ | DFDC↑ |
|---|---|---|---|---|---|---|---|
| MoCo [35] | Momentum Contrast | | G | G | 71.5 | 61.2 | 60.2 |
| CPC [59] | Predictive Coding | | G | G | 66.7 | 65.3 | 67.9 |
| DPC [33] | Dense Predictive Coding | | L | G | 65.1 | 67.5 | 71.2 |
| VDIM [37] | Global-Local DIM | | L | G | 53.2 | 70.7 | 85.3 |
| AVTS [46] | Audio-Visual Contrast | ✓ | G | G | 72.1 | 64.9 | 63.1 |
| AVSlowFast [81] | AVC + Rotation | ✓ | G | G | 56.1 | 75.8 | 80.9 |
| Ours | Global-Local AVC | ✓ | G+L | G+L | **47.8** | **83.7** | **90.1** |

Table 5: Comparison of different contrastive objectives. All the results are based on our implementation. X-Modal specifies the methods that use a cross-modal contrastive objective; others define contrastive objective on the visual modality only. We also indicate whether each contrastive objective optimizes for global and/or local representations in spatial (Spa.) and temporal (Temp.) dimensions.

the same backbone (3D-ResNet18) and the same pretraining dataset (K-AV-15K), and follow the same experimental protocol. The results show that MoCo [35], which is successful for image classification tasks, falls short on lip reading and deepfake detection. This suggests that the "vanilla" global contrastive objective may not be effective at tasks that require local information We find that the contrastive objectives that explicitly optimize for *local spatial* representations – DPC [33], VDIM [37], and ours – generally perform well, suggesting the importance of local spatial contrastive objectives. We also see that AVSlowFast [81] achieves competitive performance although it does not explicitly optimize local contrastive objectives. This can be explained by the fact that, unlike all the other methods, AVSlowFast [81] and ours define two visual encoders to capture complementary temporal information ("slow" and "fast" changing temporal features). In addition to AVSlowFast, we explicitly optimize for global and local spatio-temporal representation via space-time factorization; the strong performance suggests the effectiveness of our global-local contrastive objectives.

**Importance of global-local joint contrastive objective**. To demonstrate the importance of jointly learning global-local representations, we ablate three components of our model: 1) do we need both the visual encoders $E_v^g$ and $E_v^l$? 2) do we need both the contrastive objectives $\mathcal{L}^g$ and $\mathcal{L}^l$? 3) do we need features from both the subspaces $z_v^g$ and $z_v^l$ for the downstream tasks? In Table 6 we evaluate various combinations of these. We can see that jointly optimizing both the contrastive objectives holistically improves representations in both the subspaces $z_v^g$ and $z_v^l$. What is particularly interesting is that optimizing the objectives not only helps learn the subspace that they are defined in, but it also helps learn the other subspace. For example, compared to optimizing only $\mathcal{L}^g$, optimizing it *jointly* with $\mathcal{L}^l$ helps also improve $z_v^g$ (lines a and b). We also evaluate a variant of our approach

| | V. Enc. | Obj. | Feat. | LRS2↓ | LRW↑ | DFDC↑ | UCF101↑ | HMDB51↑ |
|---|---|---|---|---|---|---|---|---|
| (a) | Both | LG ($\mathcal{L}^g$) | LG ($z_v^g$) | 70.9 | 65.3 | 67.9 | 82.3 | 57.1 |
| (c) | Both | Both | LG ($z_v^g$) | 47.6 | 86.8 | 92.6 | 89.2 | 59.9 |
| (c) | Both | GL ($\mathcal{L}^l$) | GL ($z_v^l$) | 68.6 | 65.1 | 70.3 | 82.1 | 55.6 |
| (d) | GL ($E_v^l$) | Both | GL ($z_v^l$) | 50.8 | 81.2 | 89.7 | 83.6 | 57.3 |
| (e) | Both | Both | GL ($z_v^l$) | 46.5 | 88.9 | 95.9 | 88.5 | 58.3 |
| (f) | Both | Both | Both | **45.1** | **89.2** | **96.7** | **90.1** | **61.3** |

Table 6: The roles of global and local information on different benchmarks. **LG** means spatially-local/temporally-global and **GL** means spatially-global/temporally-local. **V. Enc.**: visual encoder setup, **Obj.**: contrastive objective used during pretraining, **Feat.**: features used in downstream tasks.

| Method | LRS2↓ | LRW↑ | DFDC↑ | UCF101↑ | HMDB51↑ |
|---|---|---|---|---|---|
| Avg Pooling & Single Pos. Pair | 40.4 | 79.2 | 88.9 | 87.8 | 56.3 |
| No Pooling & Multiple Pos. Pairs | 47.8(↑ 7.4) | 83.7(↑ 4.5) | 90.1(↑ 1.2) | 88.1(↑ 0.3) | 56.8(↑ 0.5) |

Table 7: Comparison of different methods to handle multiple positive audio-visual pairs in $\mathcal{L}_l$.

that uses only the S-global/T-local encoder $E_v^l$ (line d). Pretraining this with both the objectives improves over our full model pretrained with only $\mathcal{L}^l$ (line c), again suggesting the effectiveness of our global-local joint contrastive objectives. Our full model consistently outperforms all the other variants, demonstrating the importance of all three components (encoders, objectives, and features).

**Handling multiple positive pairs in** $\mathcal{L}_l$ (Eq. 2). In our S-global/T-local contrastive objective, we handle multiple positive audio-visual pairs (due to a higher audio sampling rate) by summing up the scores of all positive pairs. Here, we validate the effectiveness of this approach by comparing it to an alternative that performs an average temporal pooling over audio features in each local window and uses the vanilla contrastive loss [16] over the synchronized audio and visual features. Table 7 shows that performance of this alternative approach drops significantly on tasks that require local information (LRS2, LRW, DFDC), while for classification tasks both approaches achieve comparable results. This suggests the importance of our formulation specifically on capturing local information.

**Audio window size.** In our implementation, each video frame covers one-tenth of a second and roughly maps to three audio slices ($M = 3$). We therefore fix $M = 3$ to use all available audio slices without overlap between windows; here, we vary $M \in \{1, 3, 5, 7\}$ to see how that affects the performance. Table 9 shows that $M = \{3, 5\}$ is ideal. $M = 1$ leads to the worst performance due to information loss (we drop two audio slices per window),

| M | LRS2↓ | LRW↑ | DFDC↑ | UCF↑ | HMDB↑ |
|---|---|---|---|---|---|
| 1 | 49.8 | 87.1 | 95.0 | 87.8 | 57.9 |
| 3 | 45.1 | **89.2** | **96.7** | **90.1** | 61.3 |
| 5 | **44.9** | 89.0 | 96.5 | 89.5 | **61.6** |
| 7 | 46.8 | 88.1 | 95.1 | 88.3 | 59.0 |

Table 9: Comparison of different window sizes ($M$) for audio-visual temporal matching.

while $M > 5$ starts degrading performance due to the increased noise in audio-visual correspondence (each window overlaps with others).

**AV spatial attention** (Sec. 3.3). The learned audio-visual spatial attention map can highlight discriminative face regions useful for lip reading. In Table 8 we demonstrate the quality of our attention maps by replacing lip/face bounding boxes typically used in lip reading and deepfake detection with our attention map. We note that all SOTA approaches extract features from lip/face cropped regions using off-the-shelf detectors (which require substantial supervision on their own). First, as a baseline we evaluate a variant of our approach that is trained directly on full frames without using attention maps or lip/face detectors ("Ours/Full Frame"); the performance drops significantly on all three benchmarks. Next, we extract features from the entire frame (no lip/face cropping) and use our attention map to pool the features spatially. Note that the purpose of this experiment is to evaluate the quality of attention maps; we use audio signal just to obtain attention maps and discard it for lipreading/deepfake detection. The results ("Ours/Attention") show that it achieves results comparable to our best setting ("Ours/Crop"), which extracts features from the cropped lip/face region similar to the SOTA approaches. Notably, the attention-based approach outperforms SOTA on LRW and DFDC *even without relying on* lip/face region detectors, demonstrating the effectiveness.

| Task | SOTA Results | | Ours/Full Frame | Ours/Attention | Ours/Crop |
|---|---|---|---|---|---|
| LRS2↓ | TM-seq2seq [2] | 49.8 | 71.9 | 51.2 | **45.1** / Lip Crop |
| LRW↑ | DFTN [82] | 84.1 | 62.3 | 85.1 | **89.2** / Lip Crop |
| DFDC↑ | VDIM [37] | 85.3 (V) | 68.1 (V) | 95.9 (V) | **96.7** (V) / Face Crop |
| | MDS [20] | 91.5 (V+A) | | | **97.1** (V+A) / Face Crop |

Table 8: Evaluation of the learned audio-visual spatial attention maps. "V" uses only visual sequence and "V+A" uses both visual and audio sequence for finetuning on downstream tasks.

| | | Kinetics400 | | UCF101 | | HMDB51 | | ESC50 | | LRS2 | LRW | DFDC |
|---|---|---|---|---|---|---|---|---|---|---|---|---|
| Method | Backbone | LN | FT | LN | FT | LN | FT | LN | FT | FT | FT | FT |
| AVTS [46] | MC3 | - | - | - | 89.0 | - | 61.6 | 80.6 | - | - | - | - |
| XDC [5] | R(2+1)D-18 | - | - | **91.2** | - | 61.0 | **84.8** | - | - | - | - | |
| AVID [55] | R(2+1)D-50 | - | - | - | 91.5 | - | 64.7 | **89.2** | - | - | - | - |
| GDT [61] | R(2+1)D-18 | - | - | - | 92.5 | - | 66.1 | 88.5 | - | - | - | - |
| VA [4] | R(2+1)D-18 | 55.5* | - | 83.9 | 91.5 | 60.0 | 70.1 | 85.6 | - | - | - | - |
| VA [4] | S3D-G | 59.8* | - | 84.7 | 90.1 | 60.4 | 68.2 | 86.1 | - | - | - | - |
| Ours | ResNet18 | 63.7 | 71.5 | 85.1 | **93.9** | 61.2 | 73.7 | 85.1 | **89.3** | 38.1 | 95.2 | 98.9 |

Table 10: Comparison to SOTA approaches that are pretrained on AudioSet [29]. "FT": finetuning, "LN": linear evaluation. *: evaluation on Kinetics600

**Pretraining on a large-scale dataset.** We also investigate how the pretraining scale affects the results on various downstream tasks. To this end, we pretrain our model on AudioSet [29] and evaluate it on all downstream tasks. The results are show in Table 10. We report the results on both the linear (LN) evaluation and finetuning (FT) scenarios. We followed the experimental protocol used in our paper and used the same pretrained checkpoint for all downstream scenarios. For a fair comparison, we only show the results evaluated by AudioSet pretrained models for all the other comparing approaches. As we can see, among all the SOTA approaches, "Ours" achieves the best performance. When pretraining on a large-scale dataset, i.e. AudioSet with 2 million video clips, the performance can further be improved comparing with that pretrained on medium scale dataset, e.g. Kinetics and K-AV-240k.

## 5 Conclusion

We presented a contrastive approach for learning global and local spatio-temporal video representations from audio-visual correspondence. We showed that the space-time factorization leads to an effective solution for learning global-local representations. Unlike many prior work in the video self-supervised learning literature, we expand the downstream evaluation scenarios to include both "global" and "local" tasks and demonstrate that our approach successfully transfers to various tasks including lip reading, deepfake detection, audio-visual event localization, and action/sound classification.

**Limitations.** We leverage audio-visual correspondence to compute the spatial attention map.However, when applying to downstream tasks which contains videos with no audio, such a mechanism can not be used. In addition, the attention mechanism should also be considered along the temporal dimension. For example, certain frames might play more important roles in deepfake detection, e.g. frames with obvious artifacts, or in the case of lip reading, some frames or audio slices can give more clues for the model to recognize the word through visemes and/or phonemes. How to incorporate temporal or spatio-temporal attention is also an important future work.

**Boarder Impact.** Our paper studies self-supervised pretraining, with applications to tasks involving both audio and visual signals, e.g. deepfake detection, lip reading, audio-visual event localization and audio/video classification. As one of the core machine learning problems, self-supervised pretraining can enable machine learning to work better and more efficiently with less data and/or task-specific designs. Especially, audio and video signals are two key sensory signals in many real-world scenarios. In this sense, our work have broader applications in computer vision, audio/speech, bioinformatics, and human-machine intelligence. The datasets in our study are all publicly available. But every data-driven method brings the risk of learning biases in the data. Although our approach is promising in creating more safe and real environment, i.e. deepfake detection, we encourage the deployment of our method to be done with careful consideration of sensitive applications that have ethical implications.

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
