# OpenReview forum: "Contrastive Learning of Global and Local Video Representations"
_NeurIPS.cc/2021/Conference — NeurIPS 2021 Poster_

### Official Review · Reviewer_zMzE · 2021-07-10

**Rating:** 6
**Confidence:** 4

**Summary:**

This paper proposes an audio-visual self-supervised learning approach based on two cross-modal contrastive losses that learn audio-visual representations that can generalize to both the tasks which require global semantic information and localized spatio-temporal information. Extensive experiments on 4 downstream tasks demonstrate the usefulness of the learned representations, compared to their disjoint learned counterparts.

**Ethical Concerns:**

The downstream tasks in this paper involves deepfake detection and lip-reading, which could potentially lead to ethical concerns. The Broader Impacts section in the paper has discussed and well addressed the concerns.

**Limitations And Societal Impact:**

The authors have adequately addressed the limitations and potential negative social impact of the work.

**Main Review:**

Strengths:

- The paper is generally nicely written. Existing works tend to learn either global representations or local representations, while this work aims at learning versatile representations that generalize well to both scenarios.

- The motivation/advantage of learning global and local representations jointly is well justified---learning global/local representations jointly improves the generalizability of the learned representations compared to the disjointly learned counterparts, especially when the downstream scenarios are unknown in advance.

- Good observation and analysis to construct triplets from different spatial and temporal spans to capture either global information or local information. A tailored global-local contrastive learning network is designed to realize the proposed idea.

- Extensive experiments are performed on four downstream tasks across a number of datasets to demonstrate the generality of the learned representations. The proposed self-supervised audio-visual representation learning method compares favorably to other self-supervised learning methods and supervised learning methods.

- Very nice ablation studies in Table 5-7 demonstrate the effectiveness of the key components such as 1) impact of both the local and global features, 2) MIT for local contrast, and 3) attention mechanism.

Weakness / Detailed comments:

- The paper should discuss other prior work that learns global and local representations in a similar spirit, and explicitly discuss the main contribution and difference of the proposed method. E.g. [1][2]. Otherwise, the novelty of the proposed method is not salient.

- Justify the choice of downstream tasks to evaluate the learned global-local representations. Why these four tasks, and how do they represent other potential downstream tasks, since it's claimed that the proposed method learns very general representations? Or, more tasks are needed to justify that the learned representations can generalize to various tasks that leverage spatio-temporal representation.

[1] "Actbert: Learning global-local video-text representations." In CVPR, 2020.

 [2] "Contrastive learning of global and local features for medical image segmentation with limited annotations." In NeurIPS, 2020.


################################AFTER REBUTTAL###########################
Thank the authors for the response. My initial concern was that the idea of learning global and local information isn't entirely new and prior work is not acknowledged and discussed, and their choice of downstream tasks is not well justified. In their response, the authors have provided more discussions and explanations, which generally addressed my concerns. They have also provided interesting new results on action recognition (Kinetics-400) and audio-visual event localization (AVE). Therefore, I would be happy to see this paper getting accepted and encourage the authors to include the new discussions and results in the final version.




**Time Spent Reviewing:**

4

---

> ### Author Response · Authors · 2021-08-10
> **We will discuss other prior work and better justify our choice of downstream tasks.**
>
> Thanks for the overall positive comments.
>
> **Other prior work**
>
> The reviewer correctly pointed out that the idea of learning global and local information isn't entirely new. In addition to the two references [1, 2] provided by the reviewer, there are other prior works that are motivated in a similar manner, e.g., [A, B, C]. However, our work has unique aspects as summarized below:
>
> * Some of the prior work is formulated in the supervised learning regime and/or uses components pretrained with label supervision [1, A, B, C]. In contrast, we focus on self-supervised contrastive learning, and the entire model is trained end-to-end from scratch. In general, learning local representation without label supervision is difficult due to the one-to-many relationships (Lines 40-44). In this work, we address this challenge with multiple instance learning.
>
> * Some of the existing work focus on learning image representations [2, C], where the notion of global/local information is defined over the spatial dimension only. Our work focuses on learning video representations, similar to [1, A, B].
>
> * One unique aspect of our work is that we learn video representations by defining two complementary subspaces that capture S-global / T-local information and S-local / T-global information. Our specific formulation of asymmetric spatio-temporal subspaces is novel and shown to be effective in our experiments. Different from ours, ActBERT [1] incorporates global action information (spatially-global/temporally-global) and local regional objects (spatially-local/temporally-local). LGD [A] combines spatially-local/temporally-local and spatially-global/temporally-global information, while StNet [B] combines spatially-local/temporally-local and spatially-global/temporally-local information.
>
> [1] Zhu, Linchao et al. "ActBERT: Learning global-local video-text representations." CVPR, 2020.
>
>  [2] Chaitanya, Krishna et al. "Contrastive learning of global and local features for medical image segmentation with limited annotations." NeurIPS, 2020.
>
> [A] Qiu, Zhaofan, et al. "Learning spatio-temporal representation with local and global diffusion." CVPR 2019
>
> [B] He, Dongliang, et al. "StNet: Local and global spatial-temporal modeling for action recognition." AAAI 2019
>
> [C] Cao, Bingyi et al. "Unifying deep local and global features for image search." ECCV 2020
>
>
> **Justification of downstream tasks**
>
> The "global" downstream tasks we chose -- i.e., action recognition (UCF-101, HMDB-51, and Kinetics-400; see our response to Reviewer EYTG) and sound classification (ESC-50)-- are the _de facto_ standard benchmarks for the video self-supervised learning literature. We believe these datasets adequately represent the sequence-level video instance discrimination task, and they have been well-justified in the literature.
>
> On the other hand, there are no standard benchmarks that represent the "local" downstream scenario. In this work, we chose lipreading (LRS, LRW), deepfake detection (DFDC), and audio-visual event localization (AVE; see our response to Reviewer afVc) to represent tasks that require local spatio-temporal information.
>
> While there exist other video-related tasks (e.g., flow estimation, tracking, stylization, future prediction, captioning, visual question answering, etc.), we believe that our choice of downstream tasks is appropriate for evaluating audio-visual representations and that they are also the most comprehensive -- to the best of our knowledge, none of the existing work in the video self-supervised learning literature has reported results on both global and local tasks as we did in our work.
>
> We want to mention that, following the other reviewers' suggestions, we have added new results on action recognition (Kinetics-400) and audio-visual event localization (AVE). We would be happy to run additional experiments if the reviewer has specific suggestions and/or believes we are missing critical downstream tasks; please let us know!

---

### Official Review · Reviewer_afVc · 2021-07-12

**Rating:** 7
**Confidence:** 4

**Summary:**

This paper studies self-supervised learning of audio-visual representations from video data using neural networks and a contrastive audio-visual correspondence objective.

The key novelty of this paper is that it asymmetrically applies pooling/attention to simultaneously learn two different representations: spatially-local/temporally-global, and spatially-global/temporally-local. Prior work has focused on learning only one of these two, or simply learned spatially-global/temporally-global representations.

The model is pre-trained on both Kinetics-400 as well as a mixed dataset, using a 120k video subset of Kinetics-400 with a 120k video subset of the AVSpeech dataset. A single pre-trained model can then be fine-tuned on multiple downstream tasks, including action recognition on UCF101 and HMDB51, sound classification on ESC50, lip reading on LRW and LRS2, and deepfake detection on DFDC. Qualitative analysis of sounding object localization is also performed on Kinetics-Sounds. The quantitative results are strong, in most cases outperforming previous self-supervised approaches trained under similar conditions.


**Limitations And Societal Impact:**

Yes

**Main Review:**

Strengths:

The paper presents a novel and well-motivated approach to self-supervised contrastive learning from videos

The paper contains extensive experimental results on diverse downstream tasks that demonstrate the strong efficacy of the proposed approach

The paper is well-written and easy to understand

Weaknesses:

It would have been nice to evaluate on a downstream task that explicitly evaluates segmentation/localization, as opposed to tasks that only indirectly evaluate it

Suggestions:

Line 187: phoneme transcriptions are missing a slash, e.g. /p should be /p/


**Time Spent Reviewing:**

2.5

---

> ### Author Response · Authors · 2021-08-10
> **Great suggestion! We have additional results on the localization task.**
>
> We appreciate the reviewer for the positive feedback and suggestions to evaluate ours on segmentation/localization tasks.
>
> Following the suggestion, we evaluated our approach on the audio-visual event localization task of Tian et al. [A]. To this end, we took the model pretrained on K-AV-240K and finetuned it on the AVE dataset of [A]. For a fair comparison, we followed the same protocol and evaluation metric as [A]. Without bells and whistles, we achieved 82.1% localization accuracy, which outperforms DMRFE [A] (73.3%) by a large margin. We will incorporate these evaluation results into the final version.
>
> Also, thank you for spotting the typo in L187. We will fix it.
>
> [A] Yapeng Tian, Jing Shi, Bochen Li, Zhiyao Duan, Chenliang Xu. Audio-Visual Event Localization in Unconstrained Videos. ECCV 2018

---

### Official Review · Reviewer_EYTG · 2021-07-13

**Rating:** 6
**Confidence:** 5

**Summary:**

The paper introduces a novel method for self-supervised video from audio and video. The technique uses both the temporal and spatial dimension of the audio-visual input in order to build two contrastive losses: one capturing slow changing elements in the video present in the audio and the other one capturing temporally changing elements. After learning through these two losses, authors propose a detailed downstream evaluation consisting of task where spatio-temporal granularity is relevant (deepfake detection, lip reading) and other task about global information in the video (action classification).

**Limitations And Societal Impact:**

Yes, I think the authors have address the limitations of their work as well as their possible societal impact.

**Main Review:**

Overall, I think the new ideas introduced in the paper are interesting and the evaluation section show that the resulting model has good performance for lip reading, detection and classification. I am missing some comparisons with the state-of-the-art, but overall I believe this is a good paper and should be accepted.

Strengths:
- The paper is clear and well written. The different parts of the model are well motivated and it is easy for follow and understand the reason behind the design decisions.

- The idea of introducing explicitly the notion of fine-grained spatiotemporal information in contrastive learning is interesting and novel. Authors execution of the idea is simple but effective, building the two losses consistently with their goals. In my opinion, new methods for self-supervised learning will start considering more granular information, following the direction of this paper.

- Authors ablate all the relevant decisions, which to me is interesting and shows the importance and effect of the two losses used in the model. I like how looking at Table 5 you can extract the main conclusions in the paper.

- Authors also present qualitative results of the attention between the audio and the video modality. The results are consistent with the goal of the attention layer (highlight the areas in the video with relevant audio-visual correlation).

- The model is evaluated for very different tasks (lip reading, deepfake detection and classification) which is good for the reader to understand better the properties of the model.

Weaknesses:
- In terms of evaluation regimes, I am missing linear evaluation for some of the video classification tasks. It is hard to evaluate the quality of the features by only looking at the fine-tuning performance where all the network is updated on the downstream task.

- In terms of evaluation datasets, I am missing larger action classification datasets such as Kinetics-400 (where authors only train without labels). I believe having additional evaluation performances in this datasets would help the readers understand a bit better the strengths of the features.

- In terms of training datasets, I would be curious to know how this model would scale using larger scale dataset such as AudioSet. One question to answer would be if this performance improvement is only present for smaller datasets but goes away when using more data. I think it would be good if authors can answer this question.

- Although I understand that some papers cannot be directly comparable, I think it would be worth for the authors to at least report performances for some of the models trained on different architectures or datasets. I think it is important for the reader to put the quantitative performance on perspective to all the works on the topic. Some examples of this would be reporting XDC or GDT when trained in AudioSet or reporting performance of MMV.

- Authors ablate very well all the design decisions. However, I am curious about whether it is necessary to have such a large negative pool for the S-Global / T-local loss. I wonder if authors can evaluate that by just sampling some of the negatives at each iteration.

**Time Spent Reviewing:**

4

---

> ### Author Response · Authors · 2021-08-10
> **New results on AudioSet/Kinetics-400 & clarification on negative samples**
>
> We appreciate the reviewer for the detailed and overall positive comments.
>
> **Additional experiments**
>
> Thank you for suggesting us to 1) add linear evaluation results, 2) evaluate on Kinetics-400 in the downstream scenario, 3) report results pretrained on AudioSet, and 4) compare with AudioSet-pretrained results including XDC, GDT, MMV.
>
> Following the reviewer's suggestions, we pretrained our model on AudioSet and evaluated it on 7 downstream datasets: Kinetics400, UCF101, HMDB51, ESC50, LRS, LRW, DFDC. We followed the experimental protocol used in our paper and used the same pretrained checkpoint for all downstream scenarios. The table below summarizes the results. (_Higher numbers are better, except for LRS (word error rate) where the lower is better._)
>
> |  Downstream | Finetune | Linear |
> |---|:---:|:---:|
> | Kinetics-400 | 71.5 | 63.7 |
> | UCF-101 | 93.9 | 85.1 |
> | HMDB-51 | 73.7 | 61.2 |
> | ESC-50 | 89.3 | 85.1 |
> | LRS | 38.1 | 43.2 |
> | LRW | 95.2 | 86 |
> | DFDC | 98.9 | 95.1 |
>
> We agree that these results will further strengthen our paper; we will incorporate these to the final version and also report numbers from XDC, GDT, and MMV.
>
> **Negative samples for the S-Global / T-Local loss**
>
> Perhaps we confused the reviewer with our description. What we explained in Line 141 (`Given a video with $T_v$ frames and $T_a = M × T_v$ audio slices, we consider $M × (T_v − 1)^2$ negative pairs.`) is the _total_ number of negative samples for a video sequence. For the S-Global / T-Local loss, which is defined for _each_ local visual feature $z^l_v[t]$ at time $t$, we use $M × (T_v − 1)$ negative samples, which is 90 in our case ($M=3, T_v=31$). We believe this isn't actually a large number compared to the typical setting in the self-supervised learning literature for images.

---

> > ### Comment · Reviewer_EYTG · 2021-08-23
> > **Thanks!**
> >
> > Hi,
> >
> > I would like to thank the authors for the detailed response and including the downstream evaluations I suggested. In my opinion, the authors have addressed my main concerns and the paper should be accepted to the conference.
> >
> > Best wishes,

---

> > > ### Author Response · Authors · 2021-08-23
> > > **Thank you!**
> > >
> > > We are glad to hear that all the main concerns have been addressed. Thank you! We will make sure to update our paper with the new results and clarifications.

---

### Official Review · Reviewer_jwv4 · 2021-07-16

**Rating:** 6
**Confidence:** 5

**Summary:**

The paper proposes a method to train self-supervised cross-modal embeddings that are strong in both global and local representations. This is done by optimizing two objectives -- spatially-local/temporally-global and spatially-global/temporally-local. The first is trained using contrastive learning by taking negative examples from different clips, whereas the second is trained by taking negative examples from different time in the same clip. The authors conduct a range of experiments lip reading, deepfake detection, sound classification, etc.

**Ethical Concerns:**

The paper appears to be fine in this regard.

**Limitations And Societal Impact:**

The comments on limitations and broader impact are adequate. There is no particular concern beyond generic risks in ML applications.

**Main Review:**

The paper is well motivated. The authors recognise an important problem and present an effective way of solving it.

However, the key training method (e.g. sampling method for negatives) overlap strongly with Nagrani et al. [1] which is not cited in the paper. The authors should give a clear summary of the different between this paper and theirs. I may change my rating based on this.

The experiments are very thorough and convincing. The ablation studies are good. The paper is well written.

[1] https://ieeexplore.ieee.org/document/9054057

**Time Spent Reviewing:**

2hr

---

> ### Author Response · Authors · 2021-08-10
> **Key differences to Nagrani et al.**
>
> Thanks for the positive comments.
>
> We want to highlight the key differences between our approach and Nagrani et al.:
> * Their approach is specifically designed for the task of speaker identification, while our work learns general-purpose video representation and is demonstrated on a variety of downstream tasks.
> * Their approach learns only _spatially-global_ **image** representation of human faces (by contrasting a single visual frame, chosen randomly from a short video clip, to an audio stream), while ours learns both _spatially-global/temporally-local and spatially-local/temporally-global_ **video** representations. We note that their approach could be sufficient for speaker identification (global representation of human faces can successfully capture the identity information), but as we demonstrated in our paper, it may not generalize well to tasks that require local information, such as lipreading and deepfake detection.
> * Although their approach shares a similar negative sampling strategy as ours, they share the subnetworks for both content and identity objectives; this makes sense given that they optimize for a single type (spatially-global) of visual representation. In contrast, we don't share the weights of the two visual encoders (global and local) and encourage them to capture complementary information (as described in Lines 158-159 of our paper).
>
> We will make sure to summarize these differences in our paper.

---

### Decision · Program_Chairs · 2021-09-27

**Decision:**

Accept (Poster)

**Comment:**

All the reviewers appreciated the global and local message, the clear exposition, the ablations, and the additional experiments carried out
in response to the reviews.